# Roundabout receptor 2 maintains inhibitory control of the adult midbrain

Bryan B Gore[1,2], Samara M Miller[2†], Yong Sang Jo[1,2†], Madison A Baird[2†], Mrinalini Hoon[3], Christina A Sanford[2], Avery Hunker[2], Weining Lu[4], Rachel O Wong[3], Larry S Zweifel[1,2]*

[1]Department of Psychiatry and Behavioral Sciences, University of Washington, Seattle, United States; [2]Department of Pharmacology, University of Washington, Seattle, United States; [3]Department of Biological Structure, University of Washington, Seattle, United States; [4]Department of Medicine, Renal Section, Boston University Medical Center, Boston, United States

**Abstract** The maintenance of excitatory and inhibitory balance in the brain is essential for its function. Here we find that the developmental axon guidance receptor Roundabout 2 (Robo2) is critical for the maintenance of inhibitory synapses in the adult ventral tegmental area (VTA), a brain region important for the production of the neurotransmitter dopamine. Following selective genetic inactivation of *Robo2* in the adult VTA of mice, reduced inhibitory control results in altered neural activity patterns, enhanced phasic dopamine release, behavioral hyperactivity, associative learning deficits, and a paradoxical inversion of psychostimulant responses. These behavioral phenotypes could be phenocopied by selective inactivation of synaptic transmission from local GABAergic neurons of the VTA, demonstrating an important function for Robo2 in regulating the excitatory and inhibitory balance of the adult brain.

*For correspondence: larryz@uw.edu

†These authors contributed equally to this work

**Competing interests:** The authors declare that no competing interests exist.

## Introduction

The midbrain dopamine system, consisting of the VTA and substantia nigra pars compacta (SNc), is essential for motor function, motivation, reward, learning, and memory. Alterations in the activity patterns of dopamine neurons have been proposed as a key contributor to mental illness (*Grace, 1991*). In addition to dopamine, accumulating evidence points to an essential balance of excitatory and inhibitory neurotransmitter systems in the brain to allow for proper function. Shifts in this balance are increasingly linked to a variety of mental disorders including schizophrenia (*Eichler and Meier, 2008*) and autism (*Nelson and Valakh, 2015*). The VTA dopamine system is also broadly implicated in addiction where changes in inhibitory and excitatory strength are proposed to underlie drug-seeking behavior (*Chen et al., 2010*). The cellular mechanisms that maintain the excitatory and inhibitory synaptic control of the adult VTA are not well known, but resolving these processes has important implications for resolving the molecular regulation of circuit connectivity.

To identify additional genes that may regulate synaptic connectivity in the midbrain dopamine system, we surveyed the Allen Institute mouse brain expression atlas (*Lein et al., 2007*) for genes with enriched or partially enriched expression in the VTA/SNc that had previously defined roles in axonal pathfinding, synaptogenesis, or plasticity, and have linkage to mental illness. One of the genes that we identified encodes for the axon guidance receptor *Robo2*. *Robo2* was first discovered as one of the four mammalian homologs to the *Drosophila Roundabout* gene (*Robo*), which was named because of the unique axonal recrossing pattern generated at the CNS midline (*Seeger et al., 1993*; *Kidd et al., 1998*; *Brose et al., 1999*; *Simpson et al., 2000*). Further studies found that Robo family members perform additional axon guidance functions to regulate brain

**eLife digest** Although no two people are alike, we all share the same basic brain structure. This similarity arises because the same developmental program takes place in every human embryo. Specific genes are activated in a designated sequence to generate the structure of a typical human brain. But what happens to these genes when development is complete – do they remain active in the adult brain?

A gene known as *Robo2* encodes a protein that helps neurons find their way through the developing brain. Many of these neurons will ultimately form part of the brain's reward system. This is a network of brain regions that communicate with one another using a chemical called dopamine. The reward system contributes to motivation, learning and memory, and also underlies drug addiction. In certain mental illnesses such as Parkinson's disease and schizophrenia, the dopamine-producing neurons in the reward system work incorrectly or die.

To find out whether *Robo2* is active in the mature nervous system, Gore et al. used genetic techniques to selectively remove the gene from the reward system of adult mice. Doing so reduced the ability of the dopamine neurons within the reward system to inhibit one another, which in turn increased their activity. This changed the behavior of the mice, making them hyperactive and less able to learn and remember. Cocaine makes normal mice more active; however, mice that lacked the *Robo2* gene became less active when given cocaine.

Overall, the work of Gore et al. suggests that developmental axon guidance genes remain important in the adult brain. Studying developmental genes such as *Robo2* may therefore open up new treatment possibilities for a number of mental illnesses and brain disorders.

wiring broadly throughout the nervous system (*Ypsilanti and Chedotal, 2014*), including midbrain dopamine neurons (*Dugan et al., 2011*). Additional studies expanded the essential functions of *Robo2* to include cell migration, synaptogenesis, synaptic plasticity, neuronal survival, and dendritic patterning during prenatal and early postnatal development (*Shen and Cowan, 2010*; *Gibson et al., 2014*; *Koropouli and Kolodkin, 2014*; *Ypsilanti and Chedotal, 2014*).

Previous studies demonstrate that the neuronal axon guidance receptor deleted in colorectal cancer (DCC) is also highly expressed in the adult dopamine neurons, is dynamically regulated by amphetamine (*Yetnikoff et al., 2007*), and mice with haploinsufficiency for DCC demonstrate blunted responses to the psychomotor activating effects of amphetamine (*Flores et al., 2005*). Consistent with the continued function of DCC in the adult nervous system, it has recently been demonstrated that DCC regulates excitatory synaptic transmission in the adult hippocampus (*Horn et al., 2013*). These findings are consistent with a role for adult expression of neuronal pathfinding genes in the regulation of the adult dopamine system.

During early development of the spinal cord, Robo and DCC signaling have opposing roles in commissural axon guidance, with DCC activation promoting axon growth toward the midline and Robo serving as a chemorepellant to prevent recrossing (*Tessier-Lavigne and Goodman, 1996*). Robo receptors are activated upon binding chemorepellants of the Slit family, Slit1, Slit2, and Slit3 (*Tessier-Lavigne and Goodman, 1996*), which cause growth cone collapse through cytoskeletal remodeling via Rho-family GTPase signaling (*Wong et al., 2001*). All three genes for the Slit ligands and the receptors *Robo1* and *Robo2* (*Marillat et al., 2002*) are expressed in the adult midbrain, including the VTA and SNc; however, the function of these ligands and receptors in this context and their cell-specific requirements has yet to be established. Of further interest, genes involved in *Robo* signaling have been linked to schizophrenia, autism, conduct disorder, intellectual disability, language impairment, and bipolar depression (*Anitha et al., 2008*; *Potkin et al., 2009*, *2010*; *Viding et al., 2010*; *Suda et al., 2011*; *Dadds et al., 2013*; *St Pourcain et al., 2014*; *Wang et al., 2014*; *Okbay et al., 2016*).

Based on the expression of *Robo2* in the adult limbic system of the brain, its role in establishing the early wiring of the midbrain dopamine system (*Dugan et al., 2011*), and its opposing function related to netrin/DCC signaling, we hypothesized that this gene may play an important role in maintaining or modulating synaptic connectivity of the adult dopamine system. To test this hypothesis,

we conditionally inactivated the *Robo2* gene exclusively in the adult VTA of mice. We find that *Robo2* regulates the inhibitory synaptic connectivity of the VTA, and that disruption of this connectivity has profound behavioral effects including altered psychomotor activity and impaired learning. Most intriguing was a paradoxical inversion of behavioral responses to the psychostimulants cocaine and amphetamine, whereby these drugs calmed hyperactivity in mice lacking *Robo2* in the adult midbrain. These findings strongly implicate Robo2 signaling in maintaining balanced control of the VTA that is highly relevant for behavioral disorders linked to altered psychomotor control and cognition such as autism, ADHD, addiction, and schizophrenia.

## Results

### Robo2 in the adult VTA regulates inhibitory synaptic transmission

Analysis of the Allen Institute mouse brain expression atlas (*Lein et al., 2007*) reveals that *Robo2* is broadly expressed in the adult brain with the highest expression in the ventral midbrain, hippocampus and cerebellum, similar to previous reports (*Marillat et al., 2002*) (*Figure 1—figure supplement 1A*). To confirm the expression of *Robo2* in the VTA, we performed immunostaining for Robo2 and tyrosine hydroxylase (TH, a dopamine neuron marker) on brain sections from adult mice (>8 weeks old). Robo2 expression was observed throughout the VTA, largely localized to TH-positive neurons, though several TH-negative neurons were observed to also express Robo2 (TH+Robo2+ = 75.28 ± 0.56%, TH+Robo2− = 7.81 ± 1.46%, and TH-Robo2+ = 16.90 ± 1.77%, n = 4) (*Figure 1A*).

To confirm *Robo2* expression in the midbrain, we performed Ribotag, a method which allows for cell-type specific enrichment of actively translating mRNAs (*Sanz et al., 2009*). Mice containing the floxed HA-tagged Ribosomal Protein L22 (*Rpl22*) gene were crossed to mice containing Cre recombinase expression under the control of the endogenous dopamine transporter locus (*Slc6a3$^{Cre/+}$*), or the vesicular GABA transporter (*Slc32a1$^{Cre/+}$*), allowing for isolation of dopamine and GABA neuron translatomes, respectively (*Figure 1—figure supplement 1D–E*). Tissue punches of the adult midbrain from double heterozygous animals (*Slc6a3$^{Cre/+}$::Rpl22$^{lox-HA/+}$*, n = 3; *Slc32a1$^{Cre/+}$::Rpl22$^{lox-HA/+}$*, n = 3) were immunoprecipitated (IP) to isolate Rpl22-HA tagged polyribosomes. TaqMan real time PCR was used to test for the relative enrichment of *Robo2* in the IP fraction relative to total mRNA (input). We found *Robo2* to have enriched expression in the IP fraction relative to the input in both mouse lines (*Figure 1—figure supplement 1F*) confirming *Robo2* expression in both dopamine and GABA neurons of the midbrain neurons.

Based on previous findings that DCC regulates excitatory synaptic strength in the adult hippocampus (*Horn et al., 2013*), we hypothesized that Robo2 may also play an important role in this process. To establish whether *Robo2* in the adult VTA plays a functional role in synaptic function, we measured spontaneous excitatory and inhibitory postsynaptic currents (sEPSCs and sIPSCs, respectively) in putative dopamine and non-dopamine neurons following *Robo2* inactivation. To inactivate *Robo2*, an adeno-associated viral vector (AAV) containing an expression cassette for a GFP-tagged Cre recombinase (AAV1-Cre-GFP) was injected into the VTA of mice with a Cre-conditional allele for *Robo2* ('mutant mice') (*Figure 1B–C* and *Figure 1—figure supplement 1B–C*). The generation of *Robo2$^{lox/lox}$* mice were described previously (*Lu et al., 2007*). In this line, loxP recognition sites are inserted to flank exon 5 of *Robo2*, and following Cre-mediated recombination, a frameshift occurs resulting in a premature stop codon in exon 6 (*Figure 1B*). We confirmed a specific reduction in Robo2 protein in cells expressing Cre-GFP but not in cells expressing the enzymatically dead version of Cre, ΔCre-GFP (*Figure 1—figure supplement 1G–H*).

Putative dopamine neurons in an acute slice preparation from AAV1-Cre-GFP::*Robo2$^{lox/lox}$* (mutant) and AAV1-ΔCre-GFP::*Robo2$^{lox/lox}$* (control) were identified by the presence of an afterhyperpolarization induced current ($I_h$) as described (*Johnson and North, 1992*) (*Figure 1—figure supplement 2A*), though this is not a definitive feature of all dopamine producing neurons (*Margolis et al., 2006*). Surprisingly, we did not observe significant changes in either sEPSC frequency or amplitude (*Figure 1—figure supplement 2B-DA-C*). In contrast, we did observe a significant reduction in sIPSC frequency in mutant mice relative to controls (*Figure 1D–E*), with no change in sIPSC amplitude (*Figure 1D and F*).

Due to expression in putative non-dopamine neurons, as well as putative dopamine neurons, Robo2 signaling might also alter the connectivity and/or physiology of these cells. To test this, we

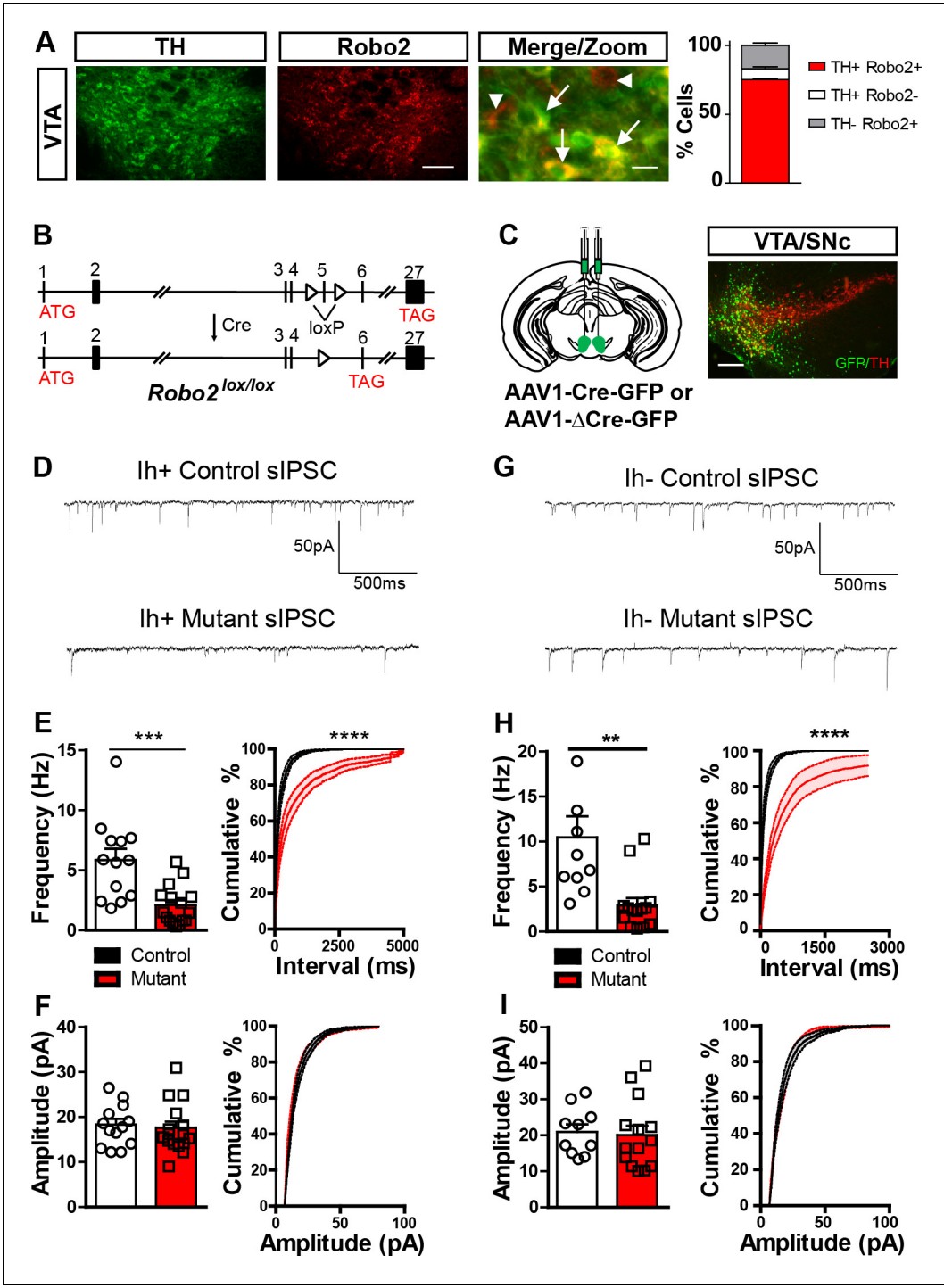

**Figure 1.** *Robo2* VTA mutants have a reduction in the frequency of sIPSCs. (A) Robo2 is expressed in the VTA, predominantly in dopamine neurons (TH+ cells, arrows), but also non-dopamine neurons (TH- cells, arrowheads). Scale bar = 100 μm (Robo2), 10 μm (merge/zoom). (B) The genomic structure of the *Robo2^{lox/lox}* allele before and after Cre recombination, adapted from *Lu et al. (2007)*. (C) Schematic of strategy to virally inactivate *Robo2* in the VTA (left). Representative image (right) showing expression of Cre-GFP and TH in the VTA. Scale bar = 100 μm. (D) sIPSC representative traces for control and mutant $I_h$ positive cell. (E) sIPSC frequency (left) and cumulative distribution plot (right) for control (n = 13) and mutant (n = 16) $I_h$ positive cells. Unpaired t-test, $t_{(27)}$ = 3.944, ***p<0.001. Two-way repeated measures ANOVA, genotype x time interaction, $F_{(27,100)}$ = 17.30, ****p<0.0001. (F) sIPSC amplitude (left) and cumulative distribution plot (right) for control and mutant $I_h$ positive cells. (G) sIPSC representative trace for control and mutant $I_h$ negative cell. (H) sIPSC frequency (left) and cumulative distribution

*Figure 1 continued on next page*

*Figure 1 continued*
plot (right) for control (n = 10) and mutant (n = 14) $I_h$ negative cells. Unpaired t-test, $t_{(22)}$ = 3.459, **p<0.01. Two-way repeated measures ANOVA, genotype x time interaction, $F_{(22,100)}$=11.40, ****p<0.0001. (I) sIPSC amplitude (left) and cumulative distribution plot (right) for control and mutant $I_h$ negative cells. Bars represent mean ± SEM.

The following figure supplements are available for figure 1:

**Figure supplement 1.** *Robo2* VTA mutants.

**Figure supplement 2.** *Robo2* VTA mutants have normal sEPSCs.

**Figure supplement 3.** *Robo2* VTA mutants.

recorded sIPSCs and sEPSCs on putative non-dopamine neurons in the VTA ($I_h$-negative; *Figure 1—figure supplement 2E*). Similar to putative dopamine neurons, sIPSC frequency was reduced in $I_h$-negative neurons and sIPSC amplitude was not altered (*Figure 1G–I*). We also did not observe differences in sEPSCs in these neurons (*Figure 1—figure supplement 2F-HD-F*).

A reduction in sIPSC frequency following *Robo2* inactivation is consistent with a reduction in inhibitory synaptic input to midbrain neurons. Ultrastructural analysis of synaptic contacts using transmission electron microscopy did not reveal gross morphological differences in either symmetrical (inhibitory) or asymmetrical (excitatory) synapses (*Figure 1—figure supplement 3A*), nor the number of observable synapses (*Figure 1—figure supplement 3B*); indicating that the reduction in sIPSC frequency is not due to a gross loss of inhibitory synaptic connectivity. Additionally, we did not observe differences in the inactivation kinetics of sIPSCs suggesting that the subunit composition of the postsynaptic $GABA_A$ receptors is not altered (*Figure 1—figure supplement 3C–F*).

## *Robo2* inactivation in the adult VTA enhances phasic dopamine

GABAergic synaptic transmission potently regulates dopamine neuron activity patterns (*Paladini and Tepper, 1999*). To determine whether altered inhibitory synaptic inputs to dopamine neurons alters their activity in vivo, we recorded single-unit activity from the VTA of control and mutant mice (*Figure 2—figure supplement 1A*). Putative dopamine neurons were identified by end of session sensitivity to the D2 agonist quinpirole that could be reversed with the D2 antagonist eticlopride (*Figure 2A–B*). No differences in quinpirole sensitivity were observed between the two groups (*Figure 2—figure supplement 1D–F*). We observed no difference between controls and *Robo2* VTA mutants in the basal firing rate (*Figure 2C*), nor did we detect significant differences in the number of burst firing, or phasic events (*Figure 2D*). In contrast, we did observe a significant increase in the duration of burst events (*Figure 2E*) and the number of spikes per burst (*Figure 2F*), consistent with a reduced inhibitory termination of burst epochs.

To determine whether activity patterns of putative non-dopamine neurons are altered in vivo, we analyzed firing of quinpirole insensitive neurons recorded on the same tetrodes as quinpirole sensitive neurons. Similar to putative dopamine neurons, quinpirole insensitive neurons did not show changes in overall firing rate (*Figure 2G*). However, these cells did show a significant change in the skew of their interspike-interval distribution (ISI, *Figure 2H–I*). Consistent with this change in ISI distribution, quinpirole insensitive neurons showed a reduction in the coefficient of variation of the ISI (CV-ISI, *Figure 2J*), indicative of a reduced number of inhibitory pauses.

We next sought to determine whether changes in dopamine activity patterns result in altered dopamine release. To test this, we performed fast scan cyclic voltammetry (FSCV) to monitor dopamine release in the nucleus accumbens (NAc) in *Robo2* mutant mice. Dopamine release was measured in response to increasing stimulus intensity delivered to an excitatory afferent input, the pedunculopontine tegmental area (PPTg) in urethane-anesthetized animals (*Figure 3A–B*; *Figure 3—figure supplement 1A*) as previously described (*Soden et al., 2013*). At lower stimulus intensities, dopamine release was significantly elevated in *Robo2* mutant mice compared to controls (*Figure 3C–H*). We did not observe significant differences in dopamine release evoked by direct stimulation of the medial forebrain bundle (MFB; *Figure 3—figure supplement 1E–J*), suggesting that dopamine transporter function is not altered in *Robo2* mutant mice. Consistent with a lack of

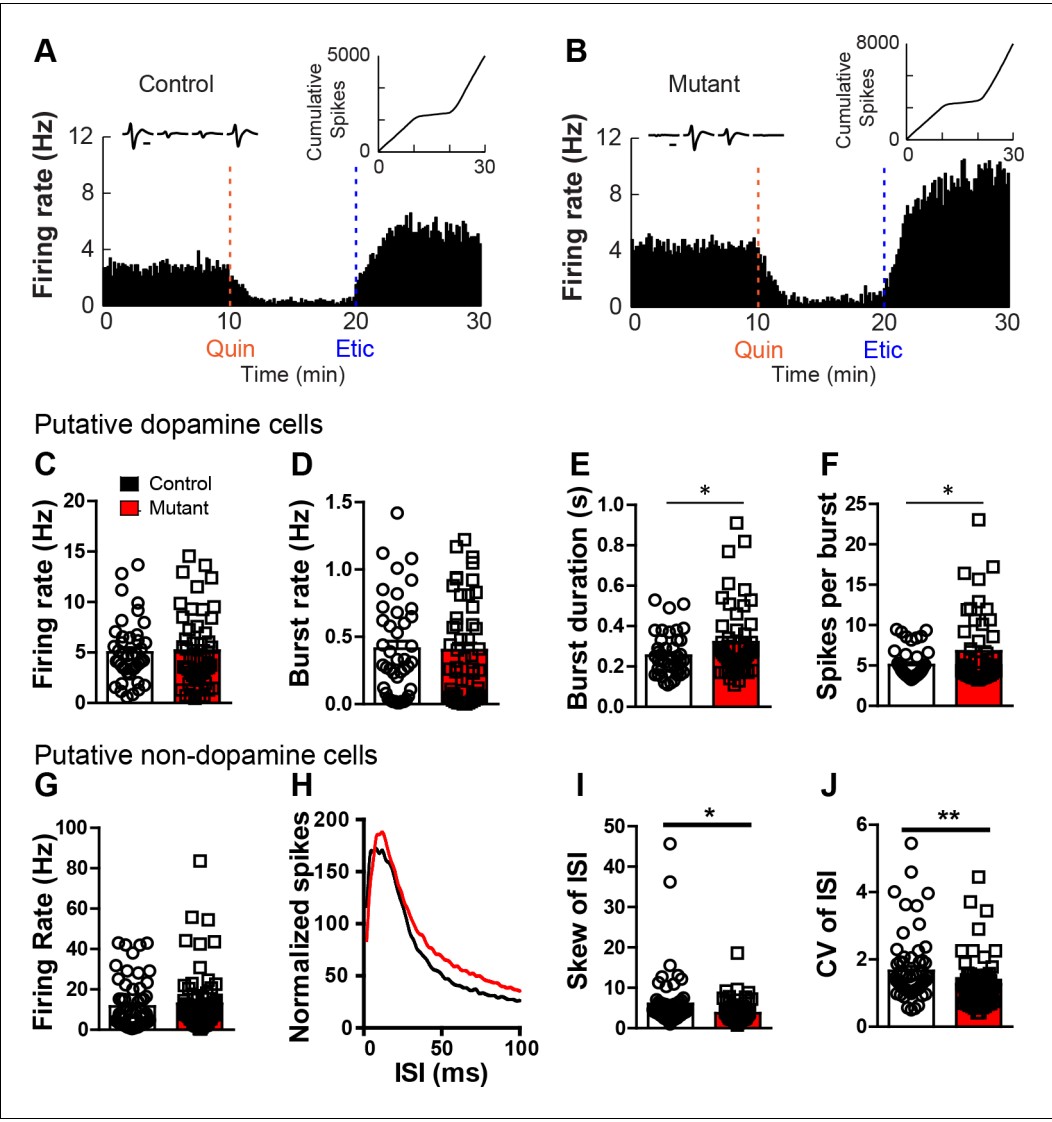

**Figure 2.** *Robo2* VTA mutants show increased burst activity. (**A,B**) Firing rate histogram of a representative cell displaying inhibition to quinpirole that is reversed with eticlopride in control (**A**) and mutant animal (**B**). (Left inset) Waveform of represented cell on each wire of the tetrode. (Right inset) Cumulative spikes of represented cell across time (min). Scale bar = 0.5 ms. (**C**) Firing rate of dopamine cells in controls (n = 42) and mutants (n = 50) is unchanged. (**D**) Burst rate of dopamine cells. (**E**) Burst duration of dopamine cells is longer in mutant cells. Unpaired t-test, $t_{(90)}$ = 2.17, *p<0.05. (**F**) Spikes per burst in dopamine cells are increased in mutants. Unpaired t-test, $t_{(90)}$ = 2.33, *p<0.05. (**G**) Firing rate of non-dopamine cells during in vivo electrophysiological recordings in controls (n = 61) and mutants (n = 71). (**H**) Total normalized distribution of the inter-spike interval (ISI) in controls and mutants. (**I**) Skew of the ISI in non-dopamine cells is reduced in *Robo2* VTA mutants. Unpaired t-test, $t_{(130)}$ = 2.42, *p<0.05. (**J**) Coefficient of variation of the ISI in non-dopamine cells is reduced in mutants. Unpaired t-test, $t_{(130)}$ = 2.66, **p<0.01. Bar represent mean ± SEM.

The following figure supplement is available for figure 2:

**Figure supplement 1.** in vivo electrophysiology.

observed differences in the overall firing rate of dopamine neurons (*Figure 2C*), we did not detect differences in tonic dopamine release as measured by microdialysis (*Figure 3—figure supplement 1K–L*).

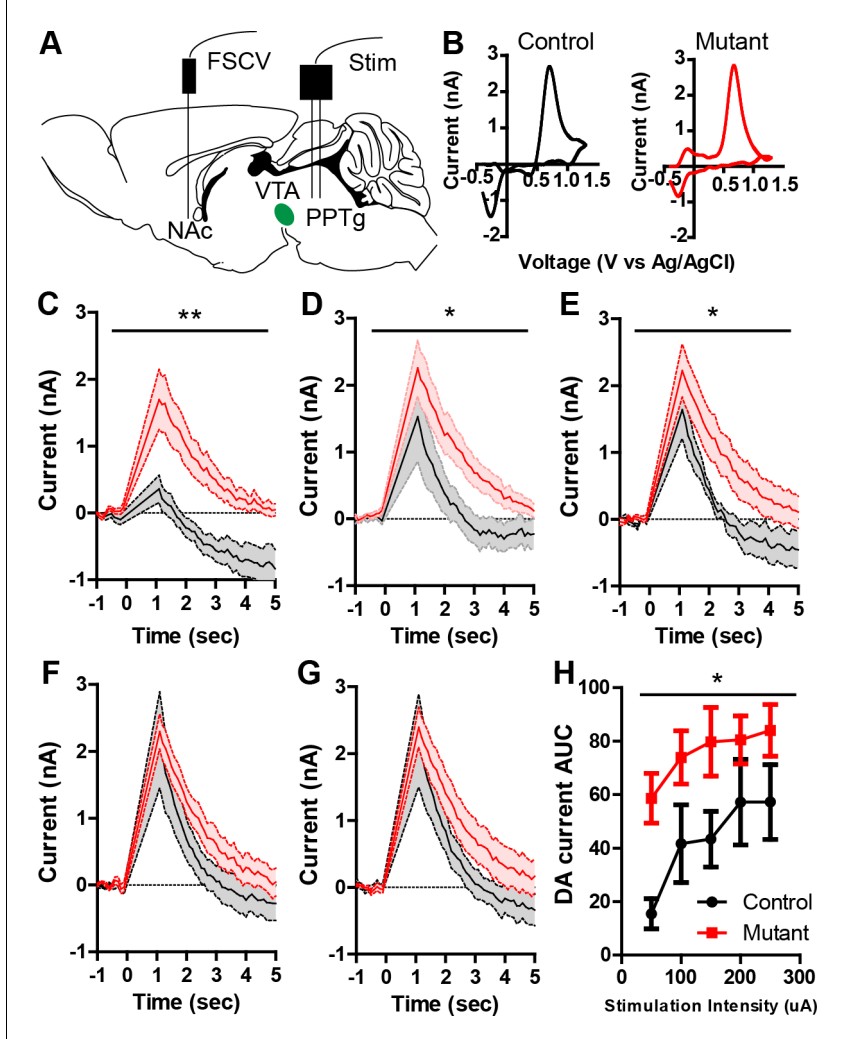

**Figure 3.** *Robo2* VTA mutants have increased phasic dopamine release. (**A**) Schematic of experimental design showing stimulation of the PPTg and FSCV recording in the NAc. (**B**) Representative voltammogram of dopamine recorded in the NAc in control (left) and mutant (right). (**C–G**) Average dopamine oxidation current recorded in NAc after PPTg stimulation is increased in mutants (n = 6) compared to controls (n = 5). (**C**) 50 µA: Two-way repeated measures ANOVA, effect of genotype, $F_{(1,9)}$ = 12.75, **p<0.01. (**D**) 100 µA: Two-way repeated measures ANOVA, effect of genotype, $F_{(1,9)}$ = 5.84, *p<0.05. (**E**) 150 µA: Two-way repeated measures ANOVA, effect of genotype, $F_{(1,9)}$ = 8.65, *p<0.05. (**F**) 200 µA. (**G**) 250 µA. (**H**) Area under the curve of the dopamine current across all stimulation intensities. Two-way repeated measures ANOVA, effect of genotype, $F_{(1,9)}$ = 5.99, *p<0.05. Bar represent mean ± SEM.

The following figure supplement is available for figure 3:

**Figure supplement 1.** Similar dopamine levels were recorded after MFB stimulation or microdialysis.

## Inactivation of *Robo2* in the adult VTA causes psychomotor dysfunction

To test whether inactivation of *Robo2* in the adult VTA has a functional impact on dopamine-dependent behavioral regulation, we monitored *Robo2* mutant and control mice in a series of behavioral tests. Both AAV1-ΔCre-GFP injected *Robo2^lox/lox^* mice and AAV1-Cre-GFP injected *Robo2^+/+^* mice were used as controls. We first tested locomotor activity across the day-night cycle. *Robo2* mutants were significantly hyperactive compared to controls (***Figure 4A***), particularly during the night

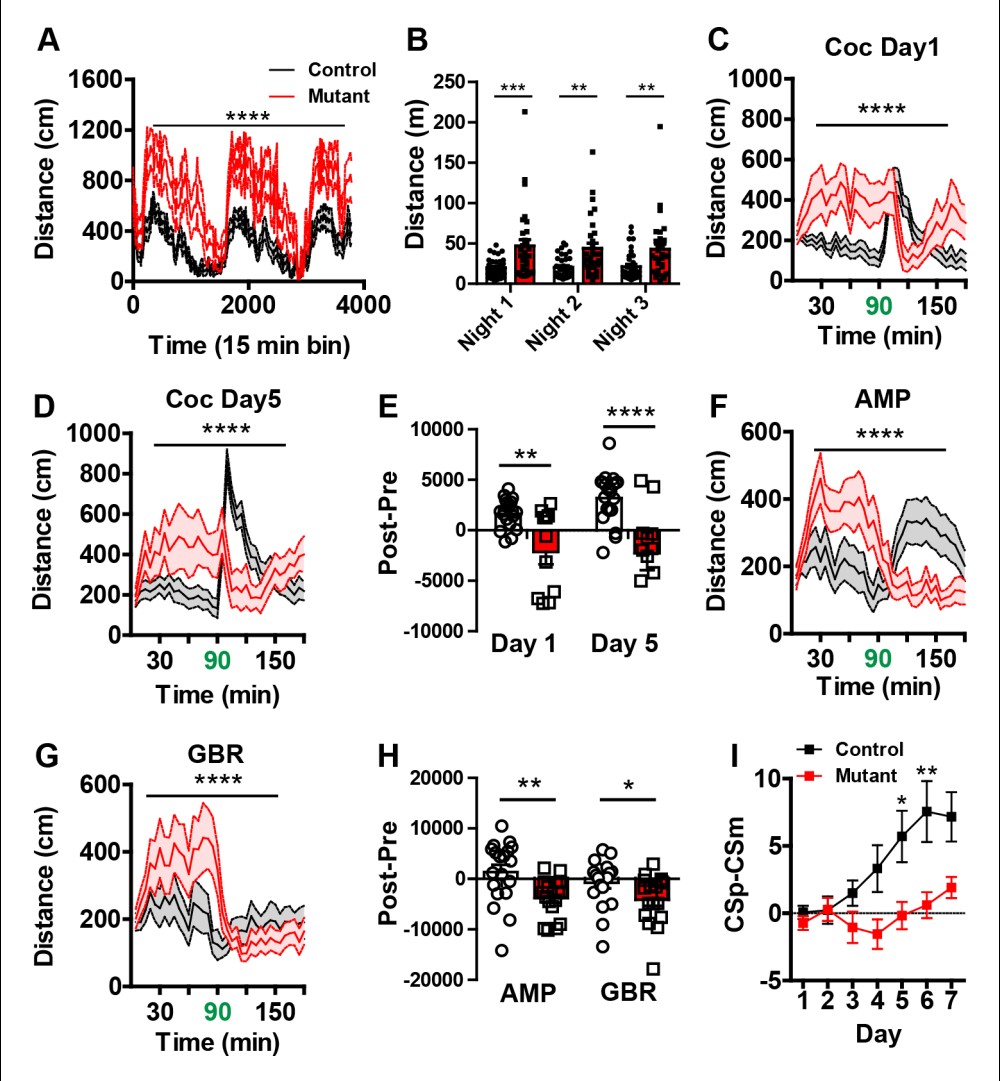

**Figure 4.** Behavioral characterization of *Robo2* VTA mutants show phenotypes in locomotion and cocaine sensitization. (**A**) Locomotion measured across three consecutive days and nights showing mutants (n = 31) are hyperactive relative to controls (n = 36). Two-way repeated measures ANOVA, genotype x time interaction, $F_{(252,16380)}$ = 2.53, ****p<0.0001. (**B**) Total distance traveled across three consecutive nights. Two-way repeated ANOVA, effect of genotype, $F_{(1,65)}$ = 12.49, ***p<0.001, Bonferroni's multiple comparison test, ***p<0.001, **p<0.01. (**C–D**) Locomotor response to cocaine (20 mg/kg) in controls (n = 20) and mutants (n = 11) on day 1 (**C**), Two-way repeated measures ANOVA, genotype x time interaction, $F_{(35,1015)}$ = 7.10, ****p<0.0001 and day 5 (**D**), Two-way repeated measures ANOVA, genotype x time interaction, $F_{(35,1015)}$ = 8.23, ****p<0.0001. (**E**) Normalized locomotor response to cocaine by subtracting 60 min pre-cocaine from 60 min post-cocaine. Two-way repeated measures ANOVA, effect of genotype, $F_{(1,29)}$ = 18.26, ***p<0.001, Bonferroni's multiple comparison test, ****p<0.0001, **p<0.01. (**F**) Locomotor response to amphetamine (2.5 mg/kg) in controls (n = 19) and mutants (n = 17). Two-way repeated measures ANOVA, genotype x time interaction, $F_{(35,1190)}$ = 6.43, ****p<0.0001. (**G**) Locomotor response to the DAT blocker, GBR-12909 (10 mg/kg) in controls (n = 19) and mutants (n = 17). Two-way repeated measures ANOVA, genotype x time interaction, $F_{(35,1190)}$ = 2.94, ****p<0.0001. (**H**) Normalized locomotor response to amphetamine and GBR-12909 by subtracting 60 min pre-drug from 60 min post-drug. AMP: Unpaired t-test, $t_{(34)}$ = 2.968 **p<0.01,. GBR: Unpaired t-test, $t_{(34)}$ = 2.061, *p<0.05. (**I**) Discrimination between a CSp and CSm is impaired in *AAV1-Cre-GFP::Robo2^{lox/lox}* animals (n = 16) compared to controls (n = 22). Two-way repeated measures ANOVA, genotype x time interaction, $F_{(6,216)}$ = 2.40, *p<0.05, Bonferroni's multiple comparison test, **p<0.01 and *p<0.05. Bars represent mean ± SEM.

The following figure supplements are available for figure 4:

*Figure 4 continued on next page*

*Figure 4 continued*

**Figure supplement 1.** *Robo2* VTA mutants have altered gait.
**Figure supplement 2.** Additional pharmacological studies on *Robo2* VTA mutant mice.

(*Figure 4B*). Consistent with altered locomotion in mutant mice, we observed reduced forelimb separation and greater print separation (*Figure 4—figure supplement 1A–F*).

Increased locomotor activity in *Robo2* mutant mice suggests these animals may have enhanced psychomotor activation. To test this, control and *Robo2* mutant mice were subjected to daily injections of cocaine (20 mg/kg) for 5 days. In control mice, we observed a characteristic locomotor sensitization to cocaine (*Figure 4C–E*). In contrast, *Robo2* mutant mice showed an inversion in the normal locomotor response to cocaine (*Figure 4C–E*), decreasing rather than increasing their activity; this pattern persisted across days. To determine whether the inverted psychomotor response to cocaine in *Robo2* mutant mice leads to an anhedonic as opposed to a hedonic state, we measured conditioned place preference (CPP) to cocaine. Although cocaine CPP was not as robust in mutant mice compared to controls (*Figure 4—figure supplement 2A*), it did not lead to an avoidance of the cocaine-paired side, suggesting that although their psychomotor response is altered, the hedonic aspect of cocaine is not significantly changed.

An inverted psychostimulant response to cocaine is remarkably similar to that reported following genetic inactivation of the dopamine transporter (*Slc6a3*-KO mice) (*Gainetdinov et al., 1999*). Analysis of *Slc6a3*-KO mice demonstrated that these animals not only have inverted responses to psychostimulants, but also a variety of monoamine transport blockers (*Gainetdinov et al., 1999*). Similar to their response to cocaine, *Robo2* mutant mice had inverted locomotor responses to amphetamine (AMP) and the DAT blocker GBR12909 (GBR; *Figure 4F–H*). *Robo2* mutant mice also had a differential response to the serotonin transport blocker fluoxetine with the mutants showing greater inhibition, but the norepinephrine transport blocker nisoxetine reduced activity equally in control and mutant mice (*Figure 4—figure supplement 2B–E*).

In addition to regulating psychomotor activation, VTA dopamine neurons play an essential role in regulating reward-related learning. To establish whether alterations in the dopamine system of *Robo2* mutant mice affect this process, we monitored their performance in a Pavlovian associative learning paradigm. *Robo2* mutant mice showed significant deficits in this task compared to control mice, with mutant mice failing to discriminate between predictive (CSplus) and non-predictive (CSminus) cues (*Figure 4I*). This impairment was not associated with a reduction in reward seeking as total head entries into the reward receptacle did not differ (*Figure 4—figure supplement 2F*).

Hyperactivity and cognitive or learning impairments associated with enhanced phasic dopamine release may be related to a psychosis-like state in mutant mice (*Grace, 1991*). To address this, we monitored pre-pulse inhibition of the acoustic startle reflex and enhanced sensitivity to the psychomimetic NMDA receptor antagonist MK-801, two metrics used to assess psychosis-related phenotypes in animal models (*Geyer, 2008*). *Robo2* mutant mice did not differ from control mice in PPI, and although they are hyperactive at baseline, they do not show increased sensitivity to MK-801. (*Figure 4—figure supplement 2G–H*).

## Behavioral dysfunction requires *Robo2* inactivation globally in the VTA

Alterations in inhibitory control of dopamine neurons, enhanced phasic dopamine release, behavioral responses to psychostimulant drugs, and impaired reward learning all point to dopaminergic phenotypes following *Robo2* inactivation in the VTA. To establish whether the observed behavioral phenotypes are the result of inactivation in dopamine producing neurons of adult mice, we crossed *Robo2^{lox/lox}* mice to mice containing an inducible Cre coupled to the estrogen receptor (iCreER) under control of the dopamine transporter promotor (*Slc6A3-iCre/ERT2*). Treatment of adult mice with the estrogen receptor agonist tamoxifen (75 mg/kg x 5 days) resulted in Cre-mediated recombination in dopamine neurons of the VTA and SNc (*Figure 5A*). Behavioral analysis of tamoxifen treated *DAT-iCreER::Robo2^{lox/lox}* mice did not result in hyperactivity, altered psychostimulant

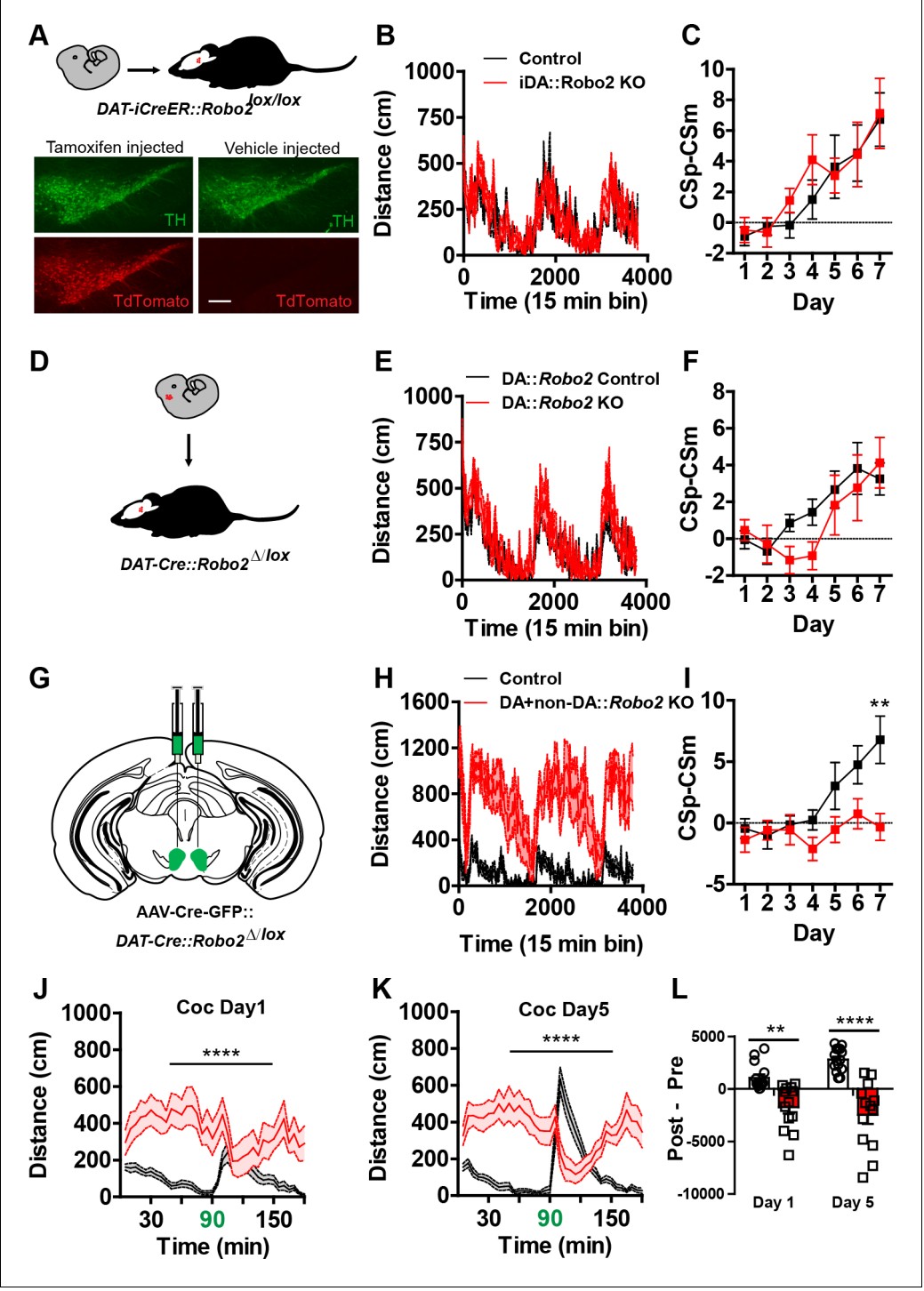

**Figure 5.** Adult or embryonic inactivation of *Robo2* in dopamine neurons does not recapitulate the *Robo2* VTA mutant phenotypes. (**A**) Schematic showing expression of tamoxifen-inducible Cre specifically in adult dopamine neurons. Representative images from *Slc6a3-iCreERT2* crossed to a Cre-dependent reporter line, *Gt(ROSA)26Sor^tm(CAG-tdTomato)Hze*, shows induction of the tdTomato reporter is specific to animals receiving tamoxifen injections. Scale bar = 100 μm. (**B**) Locomotion measured across three consecutive days and nights showing mutants (n = 12) are normal compared to control animals (n = 12). (**C**) Discrimination between CSp and CSm (left) is normal in *Slc6a3-iCreERT2::Robo2^lox/lox* animals (n = 12) compared to controls (n = 12). (**D**) Schematic showing expression of Cre in dopamine neurons starting in embryonic development and continuing into adulthood. (**E**) Locomotion measured across three consecutive days and nights showing mutants (n = 12) are normal compared to control

*Figure 5 continued on next page*

*Figure 5 continued*

animals (n = 11). (F) Discrimination between CSp and CSm (left) is normal in *Slc6a3-Cre::Robo2*$^{\Delta/lox}$ animals (n = 11) compared to controls (n = 12). (G) Schematic of strategy for injecting AAV1-Cre-GFP into the midbrain of *Slc6a3-Cre; Robo2*$^{\Delta/lox}$ animals. (H) Locomotion measured across three consecutive days and nights showing mutants (n = 13) are hyperactive relative to controls (n = 15). Two-way repeated measures ANOVA, genotype x time interaction, $F_{(252,6552)}$ = 3.17, ****p<0.0001. (I) Discrimination between a CSp and CSm is impaired in AAV1-CreGFP::*Slc6a3-Cre::Robo2*$^{\Delta/lox}$ animals (n = 6) compared to controls (n = 8). Two-way repeated measures ANOVA, genotype x time interaction, $F_{(6,72)}$ = 2.68, *p<0.05, Bonferroni's multiple comparison test, **p<0.01. (J, K) Locomotor response to cocaine (20 mg/kg) in controls (n = 15) and mutants (n = 13) on day 1 (d), Two-way repeated measures ANOVA, genotype x time interaction, $F_{(35,910)}$=6.57, ****p<0.0001 and day 5 (e), Two-way repeated measures ANOVA, genotype x time interaction, $F_{(35,910)}$= 17.03, ****p<0.0001. (L) Normalized locomotor response to cocaine by subtracting 60 min pre-cocaine from 60 min post-cocaine on day 1 and day 5, Two-way repeated measures ANOVA, genotype x time interaction, $F_{(1,26)}$= 10.97, **p<0.01, Bonferroni's multiple comparison test, ****p<0.0001, **p<0.01. Bars represent mean ± SEM.

The following figure supplement is available for figure 5:

**Figure supplement 1.** Embryonic or adult inactivation of *Robo2* does not recapitulate the *Robo2* VTA mutant phenotypes.

response to cocaine, or learning deficits relative to sham injected *Slc6A3-iCre/ERT2::Robo2*$^{lox/lox}$ mice (*Figure 5B–C*; *Figure 5—figure supplement 1A–D*).

The observed lack of behavioral effects in *Slc6A3-iCre/ERT2::Robo2*$^{lox/lox}$ mice was unexpected, but could reflect an inefficiency in *Robo2* inactivation. The conventional Cre-driver mouse line (*Slc6A3-Cre*) has been previously shown to effectively inactivate genes in virtually all dopamine producing neurons (*Engblom et al., 2008*; *Zweifel et al., 2008*). One caveat to this mouse line is that the early expression of Cre may result in developmental compensations (*Engblom et al., 2008*). Nonetheless, we generated conditional, non-inducible knockouts of *Robo2* in dopamine neurons (*Slc6A3-Cre::Robo2*$^{\Delta/lox}$ mice, *Figure 5D*). Consistent with inactivation of *Robo2*, Ribotag analysis following injection of an AAV containing Cre-dependent Rpl22HA (AAV1-FLEX-Rpl22HA) demonstrated a significant reduction in *Robo2* mRNA in dopamine neurons (*Figure 5—figure supplement 1E*); consistent with nonsense-mediated mRNA decay (*Gibson et al., 2014*). Similar to *Slc6A3-iCre/ERT2::Robo2*$^{lox/lox}$ mice, *Slc6A3-Cre::Robo2*$^{\Delta/lox}$ mice did not show hyperactivity, altered psychostimulant response to cocaine, or learning deficits relative to control mice (*DAT-iCreER::Robo2*$^{\Delta/+}$) (*Figure 5E–F*; *Figure 5—figure supplement 1F–I*).

The lack of observed effects in *Slc6A3::Robo2*$^{\Delta/lox}$ mice could be the result of developmental compensation as discussed above, and if this were true then injection of AAV1-Cre-GFP into the VTA of *Slc6A3::Robo2*$^{\Delta/lox}$ mice should not result in behavioral alterations due to an occlusion-like effect. To test this, adult *Slc6A3::Robo2*$^{\Delta/lox}$ mice were injected with AAV1-Cre-GFP into the VTA (*AAV1-Cre-GFP::Slc6A3-Cre::Robo2*$^{\Delta/lox}$ mice, *Figure 5G*). Similar to injection of AAV1-Cre-GFP into the VTA of *Robo2*$^{lox/lox}$ mice, *AAV1-Cre-GFP::Slc6A3-Cre::Robo2*$^{\Delta/lox}$ mice were hyperactive, had inverted cocaine responses, and deficits in Pavlovian associative learning (*Figure 5H–L*; *Figure 5—figure supplement 1J–K*). These results indicate that *Robo2* inactivation is required in non-dopamine producing neurons to induce the observed behavioral phenotypes.

## Loss of local GABAergic control of the VTA evokes psychomotor dysfunction

Changes in the inhibitory control of both dopamine and non-dopamine neurons in the VTA following *Robo2* inactivation suggests that local GABAergic control of the VTA is responsible for the observed behavioral alterations. To test this, we selectively silenced GABAergic neurons in the VTA through conditional expression of the light-chain of tetanus toxin fused to GFP (GFP-TetX) to block synaptic transmission (*Kim et al., 2009*). An AAV1 vector containing a Cre-dependent expression cassette for GFP-TetX (*AAV1-FLEX-GFP-TetX*, [*Han et al., 2015*]) was injected into the VTA of mice expressing Cre under the control of the endogenous VGAT locus, *Slc6a32A1-Cre* mice (*Figure 6A*). Unexpectedly, inactivation of GABAergic transmission in the VTA of *Slc6a32A1-Cre::AAV1-FLEX-GFP-TetX*

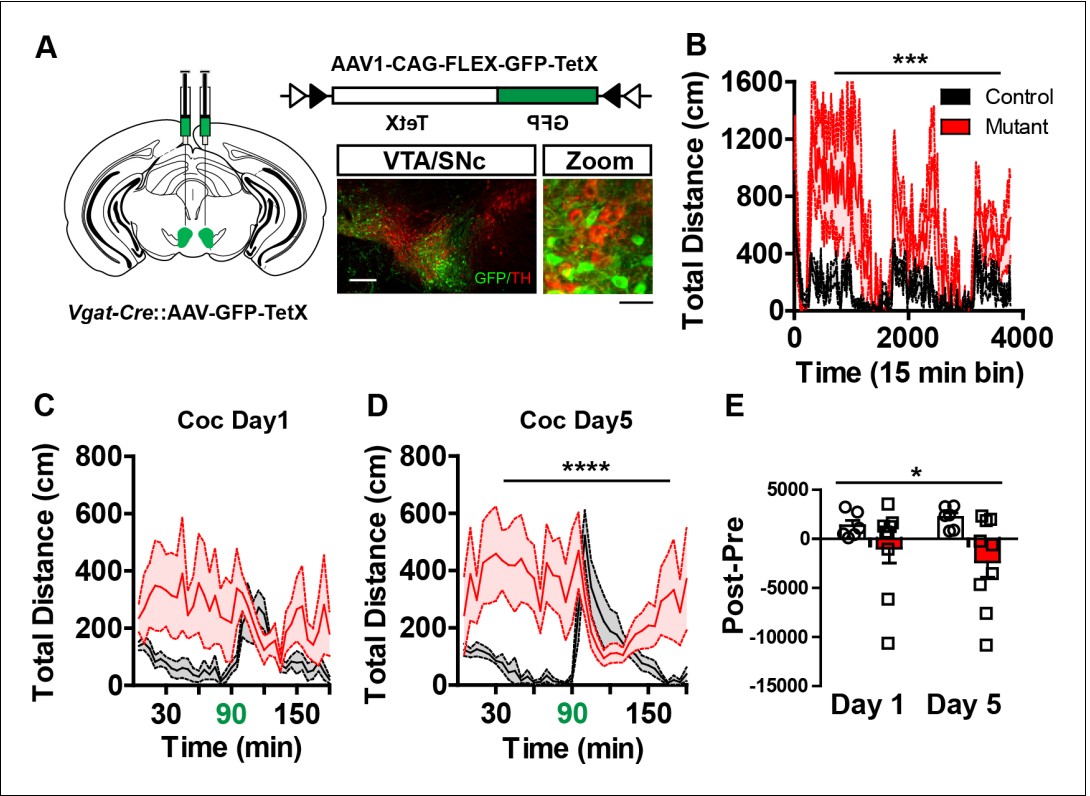

**Figure 6.** Midbrain GABA neuron inactivation recapitulates *Robo2* VTA mutant phenotypes. (**A**) Schematic of strategy for injecting AAV1-GFP-TetX into the midbrain of *Vgat-Cre* animals (left). Schematic of Cre-dependent AAV1-GFP-TetX (top). Representative image (bottom) showing viral transduction is specific to non-dopamine neurons (TH- neurons). Scale bar = 100 μm (25 μm in zoom). (**B**) Locomotion measured across three consecutive days and nights showing mutants (n = 9) are hyperactive relative to controls (n = 6). Two-way repeated measures ANOVA, genotype x time interaction, $F_{(252,3276)} = 1.35$, ***$p<0.001$. (**C,D**) Locomotor response to cocaine (20 mg/kg) in controls (n = 6) and mutants (n = 9) on day 1 (**C**), Two-way repeated measures ANOVA, genotype x time interaction, $F_{(35, 455)} = 1.34$, $p=0.0987$, and on day 5 (**D**), Two-way repeated measures ANOVA, genotype x time interaction, $F_{(35, 455)} = 2.86$, $p<0.0001$. (**E**) Normalized locomotor response to cocaine by subtracting 60 min pre-cocaine from 60 min post-cocaine on day 1 and day 5, Two-way repeated measures ANOVA, genotype x time interaction, $F_{(1,13)} = 5.35$, *$p<0.05$. Bars represent mean ± SEM.

mice resulted in rapid weight loss and compromised health in a number of animals (11 out of 20 fell below 80% of their pre-surgery bodyweight and were removed from the study) that could not be further assessed for behavioral changes. Of the remaining *Slc6a32A1-Cre::AAV1-FLEX-GFP-TetX* mice (9 out of 20) that did not show dramatic weight loss, possibly due to a reduced efficiency of GABA inactivation, we observed hyperactivity and inverted locomotor responses to cocaine (*Figure 6B–E*). Post-hoc histological analysis showed robust viral expression within the VTA, so it is unclear why the surviving animals were less affected; however, behaviorally testing the survivors possibly caused a selection bias that underestimated the severity of the phenotype. These findings suggest that reduced GABAergic control of the VTA following *Robo2* inactivation is likely the cause of hyperactivity and altered behavioral responses to psychostimulants.

## Discussion

Our results provide the first demonstration that *Robo2* has a critical function beyond early development of the nervous system by potently regulating GABAergic synapses. The observed behavioral phenotype of inverted psychostimulant responses is a rare genetic finding. Numerous other genetic models have been described that cause hyperactivity, but few have inverted psychostimulant

responses (*Leo and Gainetdinov, 2013*). Given the link between hyperactivity and paradoxical calming by psychostimulants in ADHD, it is attractive to think that Robo2 regulation of inhibitory control of the midbrain may play an important role in this disorder. Although selective inactivation of *Robo2* exclusively in the adult VTA is not a true genetic model of ADHD, our results strongly implicate balanced excitatory and inhibitory control of the midbrain as a likely source of ADHD-like phenotypes. Similar to inactivation of the dopamine transporter (*Gainetdinov et al., 1999*), we observed increased sensitivity to the serotonin transport blocker fluoxetine, suggesting the hyperactivity associated with loss of *Robo2* is due to a potential enhancement of the serotonin system.

We find that *Robo2* is highly expressed in the VTA in both dopamine and non-dopamine producing neurons. Given the dopamine-related nature of the phenotypes we observed, we were surprised to find that inactivation of *Robo2* exclusively in dopamine neurons was not sufficient to invoke these phenotypes, particularly given the partial enrichment of *Robo2* mRNA in these cells. Instead our data suggests that *Robo2* is required in multiple cell types for the observed behavioral changes. There are numerous cell types within the VTA, including dopamine neurons, GABAergic interneurons, GABAergic projection neurons, and glutamatergic neurons (*Barker et al., 2016*). Because the phenotype emerged only when dopamine and non-dopamine neurons were genetically targeted, it is possible that *Robo2* is required globally within the VTA. Unfortunately, Cre driver lines that allow for the inducible expression of Cre recombinase only in GABA or glutamate neurons of the VTA do not currently exist that would allow us to test this in greater detail. However, given that many of the observed behavioral deficits are dopamine-dependent and the genetic manipulation alters synaptic physiology in both dopamine and non-dopamine neurons, it is highly likely that the observed deficits are the results of a global change in GABAergic connectivity in the VTA. Consistent with the function of GABA in this process, reducing synaptic release from midbrain GABA neurons recapitulated the adult VTA *Robo2* mutant phenotype, though the effects were much more severe overall. These findings point to a key role of GABAergic transmission in the VTA for behavioral hyperactivity and learning deficits. Given the high degree of specificity for partial GABA agonists in targeting GABA receptors with specific subunit compositions (*Hoestgaard-Jensen et al., 2014*), drugs targeting specific GABA$_A$ receptors in the VTA may be a promising therapeutic avenue for multiple disorders.

Additional studies are required to understand the molecular mechanisms whereby Robo2 signaling controls inhibitory synapses in the VTA. The reduced frequency of sIPSCs following Robo2 inactivation is consistent with a requirement for *Robo2* in presynaptic GABA release. It was recently shown that Robo receptors interact with A-kinase anchoring proteins (AKAPs) (*Samelson et al., 2015*), placing Robo receptors in key scaffolding complexes. It has also been shown that AKAP 79/150 is expressed in dopamine and non-dopamine neurons in the VTA where it regulates GABAergic synaptic transmission. Intriguingly, disruption of AKAP function induces a long-term depression like effect on GABAergic synapse (*Dacher et al., 2013*). Thus, an intriguing possibility is that Robo2 receptors interact with AKAPs to stabilize inhibitory synaptic connectivity. Another, non-mutually exclusive possibility is that Robo2 interacts with Robo1 either in cis or in trans to stabilize synapses. Both Robo1 and Robo2 are expressed in the VTA, and it has been demonstrated in *Drosophila* that during commissural axon pathfinding, Robo2 interacts with Robo1 in trans to suppress the repulsive properties of Robo1 (*Evans et al., 2015*). Future exploration into these potential mechanisms will provide important insight into this process.

In addition to expression in the ventral midbrain, *Robo2* is highly expressed in other brain regions, most notably the limbic system including the prefrontal cortex, striatum, and hippocampus. Whether adult Robo2 signaling functions in a similar fashion in these regions will be important to elucidate, particularly given the linkage between Slit/Robo signaling and mental illness. Our results also suggest that Robo2 receptors may play an important role in regulating synaptic changes associated with drugs of abuse. Interestingly, during cocaine sensitization, changes in spine density in the striatum results from inhibition of the small Rho-family GTPase, *Rac1*, which is a known effector molecule for Robo signaling, and is a potent regulator of spine turnover during development (*Dietz et al., 2012*). In addition, cocaine sensitization reduces inhibitory synaptic transmission in the VTA (*Liu et al., 2005*) that may be mediated in part by Robo2 signaling. Determining how Slit/Robo signaling regulates synaptic processes in other cell types will be an important next step towards further resolving its function in the adult nervous system.

# Materials and methods

## Animals

Animals were group housed (max 5 animals/cage), maintained on a 12:12 light:dark cycle, and given ad libitum access to food and water except during food restriction when they were restricted to 85% of their ad libitum bodyweight. All behavioral experiments were performed during the light cycle. Equal numbers of C57Bl/6 male and female mice (8–12 weeks old) were used for all experiments. Animals were assigned to experimental groups to randomize for sex, age, and genotype. The experimenter was blinded to the genotype during data collection, and all data was generated from at least two independent experiments. The sample sizes were chosen based upon the predicted variability of the assay. Two to six independent experiments were performed for each behavioral test. In at least one of these experiments, the animals underwent a single behavioral test. When multiple tests were performed with the same group of animals, an additional cohort of animals had the behavioral tests reversed to counterbalance any interaction between tests; however, all pharmacology studies occurred subsequent to non-invasive behavioral tests (ie Pavlovian conditioning). The $Robo2^{lox/lox}$ mouse line was generated as described (*Lu et al., 2007*). The delta allele of $Robo2^{\Delta/lox}$ was generated by crossing the $Robo^{+/lox}$ to the *Meox2-Cre* mouse line to recombine the *Robo2 lox* allele. This approach was taken with the *Slc6a3-Cre* line to prevent mosaicism caused by potential transient Cre expression in the gametes of the *Slc6a3-Cre* line. The *Slc6a3-iCreERT2, Slc6a32A1-Cre*, and $Gt(ROSA)26Sor^{tm(CAG-tdTomato)Hze}$ mouse lines were obtained from Jackson Laboratory. To induce expression of *Slc6a3-iCreERT2*, daily injections of tamoxifen at 75 mg/kg were performed for five days (Sigma-Aldrich). All experiments were done in accordance with protocols approved by the University of Washington Animal Care and Use Committee.

## Stereotaxic viral injections

AAV1 virus was prepared as described (*Gore et al., 2013*). For stereotaxic injections into the brain, either AAV1-Cre-GFP, AAV1-ΔCre-GFP, or AAV1-FLEX-GFP-TetX was injected in the VTA (Relative to bregma: M-L = 0.5, A-P = −3.25*F, D-V = −4.5, F= (Bregma-Lambda)/4.21). For the generation of control mice, either a virus containing a partially deleted, nonfunctional version of Cre (AAV1-ΔCre-GFP) was injected into $Robo2^{lox/lox}$ animals (control for $Robo2^{lox/lox}$ genotype) or AAV-Cre-GFP was injected into wild type mice (control for viral Cre expression). Equal numbers of these two control groups were generated and showed no statistical difference, so they are presented as a single combined 'control' group. Animals were allowed to recover for 2 weeks prior to testing to allow for expression of the virus. Exclusion criteria was based upon histological examination that verified the expression of the virus in the VTA.

## Ribotag

For $Slc6a3^{Cre};Rpl22HA^{+/lox}$ and $Slc32a1^{Cre};Rpl22HA^{+/lox}$ experiments, tissue was collected from mice 8–10 weeks of age. For AAV1-FLEX-Rpl22HA injected $Slc6a3^{Cre}::Robo2^{\Delta/lox}$ injected mice, tissue was collected 3 weeks following injection to allow for sufficient HA-tagged Rpl22 incorporation. Tissue punches of the ventral midbrain were homogenized as described previously (*Sanz et al., 2009*). Homogenized tissue from individual mice were incubated with 5 µl of anti-HA antibody (Covance) coupled to 200 µl of magnetic beads (Pierce) overnight at 4°C. Following elution from magnetic beads, RNA from both immunoprecipitated (IP) samples and non-HA tagged (input) samples was obtained using the RNeasy micro kit (Qiagen) according to manufacturer's directions. Total RNA was quantified using a Ribogreen RNA assay kit (Invitrogen). cDNA was generated with Superscript IV (Invitrogen) using oligo dT primers from equal amounts of starting RNA, according to the manufacturer protocol. IP purified mRNA was converted to cDNA and analyzed via qRT-PCR. TaqMan (Applied Biosystems) primers against *Robo2 and Actb* were used to detect expression levels. Relative expression values were obtained using the comparative $C_T$ method and normalized to *Actb* levels. Fold enrichment was calculated as the IP versus input ratio and represented the amount of the transcript in the targeted cell type (IP) when compared to equal amounts of RNA from the input.

## Behavior

### Pavlovian conditioning

Mice underwent 7 days of conditioning, receiving 25 conditioned stimulus plus (CSp) and 25 conditioned stimulus minus (CSm) each day. The CSp and CSp were 10 s tones that were either 10 kHz or 20 kHz with a two minute intertrial interval, and animals groups receiving each tone were counterbalanced. The CSp auditory tone co-terminated with the delivery of food reward pellet 100%, whereas the CSm auditory tone co-terminated with the delivery of a food rewards pellet only 12.5%.

### Basal locomotion and pharmacology studies

For all locomotor studies, standard Allentown cages were inserted into the locomotion chamber, and activity was measured by infrared beam breaks (Columbus Instruments). For Day/Night locomotion, mice were provided ad libitum access to food and water. For pharmacological studies, animals were removed from their home cage and placed into the locomotion chamber. Locomotor activity was measured for 90 min before injections with a single compound. The compounds tested were quinpirole (0.2 mg/kg; Sigma-Aldrich), eticlopride (0.05 mg/kg; Sigma-Aldrich), nisoxetine (10 mg/kg; Tocris Bioscience), fluoxetine (20 mg/kg; Tocris Bioscience), AMPH (2.5 mg/kg; Sigma-Aldrich), cocaine (20 mg/kg; Sigma-Aldrich), GBR-12909 (10 mg/kg; Tocris Bioscience), and MK-801 (0.2 mg/kg; Sigma-Aldrich).

### Prepulse inhibition

Mice were tested in an acoustic startle chamber as described (*Soden et al., 2013*). Briefly, mice underwent 10 trials of different prepulse amplitudes, including 0 dB, 70 dB, 75 dB, and 80 dB, which preceded a 120 dB startle amplitude. The prepulse inhibition was calculated as the % reduction in startle at each prepulse intensity compared to no prepulse.

### Cocaine conditioned place preference

A three chamber CPP arena was used in an unbiased cocaine CPP paradigm (*Hnasko et al., 2007*). In brief, one side had vertical stripes and white alpha-dri bedding, and the other side had horizontal stripes and regular bedding. On Day 1, animals were placed in the center chamber and given 30 min to explore all three chambers. On the drug pairing days (Day 2 and Day 3), animals were injected with saline in the morning and confined for 30 min to one side. In the afternoon, cocaine was injected at 15 mg/kg and the animals were confined to the drug-paired side for 30 min. On the fourth day, the animals were allowed to freely explore all chambers. The % CPP was calculated as the % non-center time spent in the drug-paired side.

### Footprint analysis

The footprint analysis was performed as described (*Crawley, 2008*). In brief, forelimbs were dipped in green paint (Crayola) and the hindlimbs were dipped in red paint. The mice ran on a 6 × 55 cm piece of white paper elevated to a 10° incline. For each animal, three independent measures were made for each parameter comparing the centroid of the left and right forelimb or hindlimb.

## In vitro slice electrophysiology

Whole-cell recordings were made using an Axopatch 700B amplifier (Molecular Devices) with filtering at 1 KHz using 4–6 MΩ electrodes. Horizontal brain slices (200 μm) were prepared from 6 week old mice in an ice slush solution containing (in mM): 92 NMDG, 2.5 KCl, 1.25 $NaH_2PO_4$, 30 $NaHCO_3$, 20 HEPES, 25 glucose, 2 thiourea, 5 Na-ascorbate, 3 Na-pyruvate, 0.5 $CaCl_2$, 10 $MgSO_4$, pH 7.3–7.4 (*Ting et al., 2014*). Slices recovered for ~12 min in the same solution at 32°C and then were transferred to a room temperature solution including (in mM): 92 NaCl, 2.5 KCl, 1.25 $NaH_2PO_4$, 30 $NaHCO_3$, 20 HEPES, 25 glucose, 2 thiouria, 5 Na-ascorbate, 3 Na-pyruvate, 2 $CaCl_2$, 2 $MgSO_4$. Slices recovered for an additional 60 min. All solutions were continually bubbled with $O_2/CO_2$, and all recordings were made in ACSF containing (in mM): 126 NaCl, 2.5 KCl, 1.2 $NaH_2PO_4$, 1.2 $MgCl_2$ 11 D-glucose, 18 $NaHCO_3$, 2.4 $CaCl_2$ at 32°C continually perfused over slices at a rate of ~2 ml/min. $I_h$ currents were induced by 2 s hyperpolarizing voltage steps from −70 mV to −120 mV. All recorded

cells were medial and proximal to the MT, where the presence of an $I_h$ current correlates highly with dopamine neuron identity (*Margolis et al., 2006*).

For recording spontaneous EPSCs, electrodes were filled with an internal solution containing (in mM): 130 K-gluconate, 10 HEPES, 5 NaCl, 1 EGTA, 5 Mg-ATP, 0.5 Na-GTP, pH 7.3, 280 mOsm. 200 mM picrotoxin was included in the ACSF to inhibit $GABA_A$ receptor-mediated events. For recording spontaneous IPSCs, electrodes were filled with an internal solution containing (in mM): 135 KCl, 12 NaCl, 0.05 EGTA, 100 HEPES, 0.2 Mg-ATP, 0.02 Na-GTP, pH 7.3, 280 mOsm. 2 mM kynurenic acid was included in the ACSF to inhibit AMPA and NMDA receptor-mediated events. Cells were held at −60 mV, and access resistance was monitored throughout all experiments.

To calculate the Tau ($\tau_w$) for the sIPSCs, we calculated the weighted time constant, $\tau_w$, for 10–90% of the rise time by generating an average waveform for each cell using the equation, $\tau_w = (A1\ \tau_1) + (A2\ \tau_2)$, where $\tau_1$ and $\tau_2$ are the time constants of the first and second exponential functions, respectively, and $A_1$ and $A_2$ are the proportion of the amplitude of the IPSC of the first and second exponential functions, respectively as described (*Eyre et al., 2012*).

## In vivo electrophysiology

Four recording tetrodes were made from tungsten wires (25 µm in diameter; California Fine Wire Company) and mounted on a custom-built Microdrive (*Jo et al., 2013*, *Jo and Mizumori, 2016*). Each wire was gold-plated to adjust its final impedance to 200–400 KΩ (tested at 1 kHz). The microdrive was surgically implanted in one hemisphere of the VTA after bilateral virus infusion of AAV-Cre-GFP or AAV- ΔCre-GFP into the VTA. Two weeks after surgery, each mouse was placed in a rectangular box (21.6 cm × 17.8 cm) and single-unit activity was daily monitored as follows: the microdrive was connected to a preamplifier and its outputs were transferred to a RZ5 BioAmp processor (Tucker-Davis Technologies). Unit signals were filtered between 0.3 and 5 kHz, and digitized at 24 kHz. Tetrodes were advanced in 40 µm increments, up to 160 µm per day. Once stable and isolated units were found, their sensitivity to quinpirole and eticlopride was tested to identify putative dopamine cells (*Zweifel et al., 2009*). A VTA cell was classified as putative dopaminergic if it exhibited a low basal firing rate (less than 12 Hz) and showed severe inhibition by quinpirole (0.2 mg/ml, i.p.; ≥70% inhibition of the basal frequency) and subsequent recovery by eticlopride (0.5 mg/ml, i.p.; ≥70% the basal frequency). The other neurons that did not meet the criteria were considered as putative non-dopaminergic cells. VTA neurons were recorded for 20 min to determine baseline firing properties. Spikes from these single neurons were isolated by cluster analysis using Offline Sorter (Plexon). Subsequent data analyses, such as comparison of average firing rates and burst rates were performed with Matlab software (Mathworks, *Zweifel, 2017* https://github.com/zweifellab/ephys with a copy archived at https://github.com/elifesciences-publications/ephy). The onset of a burst was identified as two consecutive spikes with an interspike interval of <80 ms and its termination was defined as the an interspike interval of >160 ms (*Grace and Bunney, 1984*).

## Fast scan cyclic voltammetry

FSCV was performed using carbon fiber microelectrodes encased in fused-silica capillary tubing (Polymicro Technologies) (*Clark et al., 2010*). The carbon fiber microelectrode was positioned in the dorsomedial striatum at stereotaxic coordinates: A-P = 1.2 mm, M-L = 1.2 mm, D-V = 3.5 below the dura and a waveform was applied at a frequency of 60 Hz. After 40 min, the waveform frequency was reduced to 10 Hz for 10 min or until baseline recordings stabilized. Next, the carbon fiber microelectrode was lowered into the nucleus accumbens in 0.05 mm increments until maximum dopamine release was observed (D-V coordinates ranged from 3.60–3.98 below bregma, average 3.80). An Ag/AgCl reference electrode was placed in the contralateral hemisphere. The stimulation electrode (Plastics One) was placed above the PPTg at stereotaxic coordinates: A-P: −4.6 mm, M-L: 0.8 mm and lowered in 0.1 mm increments from 2.5 mm below the dura until optimal dopamine release was observed after 1 s stimulations of 60 Hz (average D-V coordinate: 2.7 mm below the dura). For the MFB stimulations, the stereotaxic coordinates were A-P: 2.4 mm, M-L: 1.1 mm, DV: 0.1 increments from 4 mm.

## Microdialysis-UPLC

Microdialysis-UPLC experiments were modified for mouse tissue from a protocol previously described (*Schindler et al., 2016*). In brief, a 1 mm long microdialysis probe (BASi instruments) was equilibrated in artificial cerebrospinal fluid (ACSF) containing 154.7 mM $Na^+$, 0.82 mM $Mg^{+2}$, 2.9 mM $K^+$, 132.49 mM $Cl^-$, 1.1 mM $Ca^{+2}$, and 5.9 mM D-glucose, and pH to 7.4 with $CO_2$. The probe was inserted unilaterally into the nucleus accumbens (M-L = +1.0, A-P = +1.3*F, D-V = 4.0, F=(distance between lambda and bregma)/4.21). The probe was flushed with ACSF at a rate of 2 µl/min for 90 min, followed by 1 µl/min for 30 min. Four 20 min fractions were collected. An equal volume of dialysate was mixed with an internal standard solution containing 100 pg/µl of deuterated D4 (2-(3,4-dihydroxyphenyl)ethyl-1,1,2,2-d4-amine HCL) 0.01 N perchloric acid and frozen on dry ice until run on the UPLC. For the detection of dopamine on the UPLC, serial dilutions of purified D4 dopamine samples of known concentrations were first analyzed to detect the retention time for dopamine on a Xevo TQ-S mass spectrometer. Samples for controls and *Robo2* VTA mutants were then ran in an interleaved manner to control for any column artifact. The dopamine and D4 peaks were identified and the area under the curve was calculated for each sample.

## Histology

Animals were deeply anesthetized with beuthanasia and transcardially perfused with 4% paraformaldehyde (PFA). The brain was removed and fixed overnight in 4% PFA before being cryoprotected in 30% sucrose. 30 µm frozen sections were made on a Leica cryostat and fluorescence imaging was performed on a Nikon upright microscope. For immunohistochemistry, the following antibodies were used: Tyrosine hydroxylase (rabbit polyclonal, 1:2000, Millipore, or mouse monoclonal, 1:2000, Millipore), GFP (mouse monoclonal, 1:1000, Invitrogen or chicken polyclonal, 1:1000, Abcam), Robo2 (rabbit polyclonal, 1:1000, Abcam (discontinued), and dsRed (rabbit, 1:1000, Clontech). For transmission electron microscopy, tissue was fixed overnight in 4% glutaraldehyde in 0.1 M sodium cacodylate buffer, pH 7.2, and then postfixed in 1% osmium tetroxide, washed, en bloc stained in 1% uranyl acetate, embedded in epoxy resin, and then sectioned to 80 nm.

## Data analysis

Statistical analyses were performed in Prism (GraphPad), Excel (Microsoft), and Matlab (Mathworks). For data that was normally distributed with equal variance between groups, an unpaired t-test or two-way repeated measures ANOVA was performed. For data that was not normally distributed (*Figure 1—figure supplement 2E*), a non-parametric Mann-Whitney test was performed. The sample sizes were chosen based upon previously established variability of the assay. The images were processed using either Photoshop (Adobe).

# Acknowledgements

We would like to thank the entire Zweifel lab for their critical reading of the manuscript. This work was supported by the NIH (R01-MH094536 and R01-MH104450 to LZ, EY10699 to RW, and R01DK078226 to WL).

# Additional information

## Funding

| Funder | Grant reference number | Author |
| --- | --- | --- |
| National Institutes of Health | R01-MH094536 | Larry S Zweifel |
| National Institute for Health Research | EY10699 | Rachel O Wong |
| National Institute for Health Research | R01-DK078226 | Weining Lu |
| National Institutes of Health | R01-MH104450 | Larry S Zweifel |

The funders had no role in study design, data collection and interpretation, or the decision to submit the work for publication.

## Author contributions
BBG, Conceptualization, Data curation, Formal analysis, Methodology, Writing—original draft, Writing—review and editing; SMM, YSJ, MAB, MH, CAS, AH, Data curation, Formal analysis, Methodology, Writing—original draft, Writing—review and editing; WL, ROW, Data curation, Formal analysis, Supervision, Funding acquisition, Methodology, Writing—original draft, Writing—review and editing; LSZ, Conceptualization, Data curation, Formal analysis, Supervision, Funding acquisition, Methodology, Writing—original draft, Writing—review and editing

## Author ORCIDs
Larry S Zweifel, http://orcid.org/0000-0003-3465-5331

## Ethics
Animal experimentation: All experiments were done in accordance with a protocol (4249-01) approved by the University of Washington Animal Care and Use Committee.

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
