## [Decision Letter]

Thank you for submitting your article "Roundabout receptor 2 maintains inhibitory control of the adult midbrain" for consideration by *eLife*. Your article has been favorably evaluated by a Senior Editor and three reviewers, one of whom is a member of our Board of Reviewing Editors. The following individual involved in review of your submission has agreed to reveal his identity: Garret D. Stuber (Reviewer #2).

The reviewers have discussed the reviews with one another and the Reviewing Editor has drafted this decision to help you prepare a revised submission.

Summary:

This is a very thorough paper by Gore et al., where they examine the role of Robo2 in the VTA of mice in regulating the activity of dopamine neurons and a variety of behavioral phenotypes. The main findings are that loss of Robo2 in the VTA reduces inhibitory synaptic drive in the VTA, results in enhanced phasic activity of VTA dopamine neurons (and dopamine release in the NAc), and is associated with enhanced locomotion. These mice show alterations in reward learning and also show a paradoxical reduction in locomotion in response to psychomotor stimulants. The authors also delete Robo2 selectively in dopamine neurons and show that this alone cannot recapitulate the behavioral phenotypes seen when the gene is removed non-specifically from VTA neurons. Thus, they conclude that Robo2 is essential for inhibitory synapses maintenance in the VTA in adult mice. Overall, the experiments are well performed and the paper is clearly written. The findings are also quite novel as to my knowledge previous work has not investigated how Robo2 alters the function of VTA neurons, despite the enrichment of the expression of this gene within the VTA.

Essential revisions:

1) The authors showed the difference of the phenotypes between AAV-mediated k/o in VTA versus DAT cre mediated k/o. Given that it is almost impossible to make AAV injections just within VTA, could it be possible that the unintended virus spread infected some cells that are right next to VTA (some interneurons right next to VTA, but outside of it)? The authors did state that exclusion criteria was based on histological examination. But it is unclear whether injections that have 'tails', which is almost impossible to avoid with VTA injections, are included. It is also unclear what are the criteria and how robust they are. The authors' explanation is also a plausible one: the k/o with DATcre is incomplete or there is compensation for the protein and therefore they generated δ allele of Robo. When using DAT-cre::Robo2Δ/lox mice, there is still one allele that suffers the same problems. Therefore, quantifying the expression of Robo is important, to actually show the incomplete of k/o and improvement with DAT-cre::Robo2Δ/lox (e.g. by IHC).

2) Given the above mentioned phenotype differences, Figure 6 is the key experiment of the paper. To the authors credit, they found that using VGAT-cre mediated k/o, there are phenotypes. However, this part of the experiments seemed done in a rush (only one sentence about the key results) and was complicated by the fact that 11/20 mice had weight loss phenotypes. What does it mean 'dramatic weight loss' and what is 'not dramatic weight loss'? How objective is this? With complications, the only convincing way is the rescue experiment: expression Robo in VGAT+ cells, or stimulation of VGAT+ cells.

3) Images in Figure 1 is the foundation of the work. The authors stated that 'Robo2 is expressed in the VTA, predominantly in dopamine neurons (TH^+^ cells), but also non-dopamine neurons.' This was referred to as 'yellow cells' and 'red cells' respectively. But it looks to me that there are also a lot of TH-positive neurons that do not express Robo (green cells). Given the possibility of non-dopamine cells in this area might be an important component of the circuits based on functional data, it is worth quantifying the three categories of the cells. Similarly, in Figure 1—figure supplement 1, if I understand correctly, top panels are supposedly to show that there are yellow signals around the green cells, and bottom panels are supposedly to show that there is no red around green cells. But from what I can see, there are definitely green cells that are not surrounded with red signals on the top panel. This needs to be quantified.

4) The authors present some example images of Robo2 localization with TH^+^ and TH- negative neurons. However, additional quantification of these data would be helpful. That is, is the proportion of TH^+^ and TH- neurons that express Robo2 similar or different.

5) Although the data implicate that Robo2 is expressed in GABAergic neurons in the VTA, there is no direct evidence to support this. The authors should try direct immunohistochemistry experiments to support this hypothesis. Ideally, the authors need to use several GABAergic neuronal markers. It would also be ideal if the authors could test whether other cell types in the VTA such as glutamatergic and GABAergic projection express Robo2 to exclude potential contribution of these neurons to the observed phenotype.

---

## [Author Response]

Essential revisions:

1) The authors showed the difference of the phenotypes between AAV-mediated k/o in VTA versus DAT cre mediated k/o. Given that it is almost impossible to make AAV injections just within VTA, could it be possible that the unintended virus spread infected some cells that are right next to VTA (some interneurons right next to VTA, but outside of it)? The authors did state that exclusion criteria was based on histological examination. But it is unclear whether injections that have 'tails', which is almost impossible to avoid with VTA injections, are included. It is also unclear what are the criteria and how robust they are.

We apologize for the confusion on the language for the exclusion criteria. The criteria that was used excluded any mice that did not have direct viral infusion into the VTA. No animals met these criteria, so no animals were excluded. For all animals in this study, the peak viral expression occurred within the VTA. A small volume of diluted virus was injected into the VTA to minimize spread outside the VTA. A minority of animals exhibited weak “tail” expression outside the VTA. We have now performed a quantitative analysis (n=10 animals) of the spread of the AAV1-Cre-GFP virus within the adult midbrain. We traced the viral spread throughout the rostral, medial, and caudal VTA. We also performed cell counts within these brain regions. We observed good coverage of the virus throughout the VTA with minimal spread outside the VTA.

In the instances of minor spillover outside the VTA, it spread to adjacent midbrain regions that do not express Robo2. As can be seen in the Robo2 in situ data (Figure 1—figure supplement 1), Robo2 expression in the adult midbrain is largely confined to the VTA and SNc, so cre-mediated recombination of the floxed Robo2 locus in non-Robo2 expressing cells outside the VTA and SNc should presumably have no effect. In the subset of the animals with minor viral spillover expression into the SNc, inactivating Robo2 in the SNc might have contributed to the phenotypes. However, we observed the same phenotypes in animals that selectively expressed the virus in the VTA with no expression in the SNc, which demonstrates the VTA is the key site of Robo2 expression in the adult responsible for the observed phenotypes.

The authors' explanation is also a plausible one: the k/o with DATcre is incomplete or there is compensation for the protein and therefore they generated δ allele of Robo. When using DAT-cre::Robo2Δ/lox mice, there is still one allele that suffers the same problems. Therefore, quantifying the expression of Robo is important, to actually show the incomplete of k/o and improvement with DAT-cre::Robo2Δ/lox (e.g. by IHC).

We apologize for the lack clarity regarding the use of the δ allele in the DAT-Cre::Robo2△/lox mice. We commonly use the δ allele for cell-specific inactivation in non-inducible cre-driver lines. This is based on the observation that many genes are transiently or ectopically expressed in germ cells, but not with 100% penetrance. Thus, in a mouse breeder that is DAT-Cre::Robo2lox/+ for example, some germ cells may have ectopically expressed Cre from the DAT locus, but this may not occur in all germ cells. Thus, some gametes will be Robo2△/+ and some will be Robo2lox/+ yielding offspring from the same litter with potentially different genotypes. To circumvent this, we generate a δ allele and only breed mice that are DAT-Cre:: Robo2△/+ with mice that are Robo2lox/lox. In this way we can avoid mosaicism within a given litter. To control for potential effects of heterozygosity, control mice are DAT-Cre::Robo2lox/+, Robo2 △/lox (Cre-negative), and DAT-Cre::Robo2+/lox. We did not observe any behavioral differences in these genotypes and collapsed these mice into a single control group. This has been more accurately described in the Methods section.

With regard to the IHC for Robo2, that relates specifically to this point as well as points 3, 4, and 5 below, the Robo2 rabbit polyclonal antibody we used in this study from Abcam that was rigorously validated by us and other investigators (for example see Gibson et al., 2014), has been discontinued. We have made a note of this discontinuation in our Methods section. This prevented us from performing additional experiments to complete some of the detailed analysis requested. We acquired multiple additional Robo2 antibodies, however these antibodies did not prove to be selective for Robo2 (likely cross-reacting with other Robo family members). In some cases, such as in relation to specific points 3 and 4, we had a sufficient amount of immunostaining from previous analyses to complete quantitative comparisons of Robo2 overlap with TH^+^ and TH- cells. To address outstanding questions relating to cell-specific expression of Robo2, we performed cell-specific mRNA analysis utilizing the Ribotag strategy (Sanz et al., 2009). Using this approach we were able to isolate the translatomes from DAT-Cre control and DAT-cre::Robo2Δ/lox mice and demonstrate a de-enrichment of Robo2 in dopamine producing neurons consistent with genetic inactivation and nonsense-mediate mRNA decay. See response to specific point 5 for additional details.

2) Given the above mentioned phenotype differences, Figure 6 is the key experiment of the paper. To the authors credit, they found that using VGAT-cre mediated k/o, there are phenotypes. However, this part of the experiments seemed done in a rush (only one sentence about the key results) and was complicated by the fact that 11/20 mice had weight loss phenotypes. What does it mean 'dramatic weight loss' and what is 'not dramatic weight loss'? How objective is this? With complications, the only convincing way is the rescue experiment: expression Robo in VGAT+ cells, or stimulation of VGAT+ cells.

We apologize for the confusion in our previous version of the manuscript; we have expanded our description of this section. This experiment was performed on three separate cohorts of 6-8 mice per group per cohort. Numerous mice with inactivation of the GABA neurons had to be euthanized because they exhibited “dramatic weight loss”, defined as animals dropping below 80% of their pre-surgery bodyweight. The surviving animals also lost bodyweight, but stabilized two weeks post-surgery, allowing for their behavioral characterization. All of the surviving animal had robust viral expression within the VTA, so they did not survive due to being partial or unilateral viral hits. Thus, we believe that our analysis of these mice is valid, caveats notwithstanding, since they did show robust viral expression. It is unclear why some animals were more affected than others, but possibly a compensatory mechanism occurred in the survivors. As noted above, we have added to the text additional information regarding this caveat. Importantly, this selection bias of only be able to analyze the survivors might have caused an underestimation of the phenotype.

3) Images in Figure 1 is the foundation of the work. The authors stated that 'Robo2 is expressed in the VTA, predominantly in dopamine neurons (TH^+^ cells), but also non-dopamine neurons.' This was referred to as 'yellow cells' and 'red cells' respectively. But it looks to me that there are also a lot of TH-positive neurons that do not express Robo (green cells). Given the possibility of non-dopamine cells in this area might be an important component of the circuits based on functional data, it is worth quantifying the three categories of the cells. Similarly, in Figure 1—figure supplement 1, if I understand correctly, top panels are supposedly to show that there are yellow signals around the green cells, and bottom panels are supposedly to show that there is no red around green cells. But from what I can see, there are definitely green cells that are not surrounded with red signals on the top panel. This needs to be quantified.

We appreciated the reviewers’ comments; fortunately we had a sufficient number of mice in which this staining was performed previously to do this quantitative analysis. We now have quantified the three categories of cells that include TH^+^Robo2+, TH^+^Robo2-, and TH-Robo2+ cells (Figure 1). In addition, we have performed Ribotag from dopamine and GABA neurons and demonstrate that Robo2 has enriched expression in both populations (Figure 1—figure supplement 1).

Regarding Figure 1—figure supplement 1 (now Figure 1—figure supplement 1), your understanding of the figure is correct; namely, the inactive Cre (top panel) has no effect on Robo2 expression, but in the active Cre (bottom panel), Cre-GFP cells are never found co-expressed with Robo2. However, since the AAV-Cre-GFP and AAV-∆Cre-GFP viruses are non-conditional, it will be expressed in a variety of cell types that include midbrain neurons that do not express Robo2; hence, some cells will be GFP positive but Robo2 negative. In both the top and bottom panel, these cells would appear as cells that are green but not red. We have now quantified the distribution of the three categories of cells, GFP+Robo2+, GFP-Robo2+, and GFP+Robo2- in animals receiving either the AAV1-Cre-GFP or the AAV1-∆Cre-GFP virus (Figure 1—figure supplement 1).

4) The authors present some example images of Robo2 localization with TH^+^ and TH- negative neurons. However, additional quantification of these data would be helpful. That is, is the proportion of TH^+^ and TH- neurons that express Robo2 similar or different.

As noted above for specific point #3, we have quantified the three categories of cells that include TH^+^Robo2+, TH^+^Robo2-, and TH-Robo2+ cells.

5) Although the data implicate that Robo2 is expressed in GABAergic neurons in the VTA, there is no direct evidence to support this. The authors should try direct immunohistochemistry experiments to support this hypothesis. Ideally, the authors need to use several GABAergic neuronal markers. It would also be ideal if the authors could test whether other cell types in the VTA such as glutamatergic and GABAergic projection express Robo2 to exclude potential contribution of these neurons to the observed phenotype.

As mentioned in response to specific points 1 and 3, we were not able to perform detailed immunohistochemical analyses due to the discontinuation of the validated antibody to Robo2. However, our Ribotag experiment that we have added to the manuscript (Figure 1—figure supplement 1), demonstrate *Robo2* to be enriched ~2 fold in purified translatomes from VGAT-Cre GABA neurons compared to the ventral midbrain tissue punch input mRNA. These findings, together with our Ribotag analysis of dopamine neurons correlate with the immunohistochemistry showing expression of Robo2 in both midbrain dopamine (TH^+^) and non-dopamine (TH-) neurons.